# Impact of Various Essential Oils and Plant Extracts on the Characterization of the Composite Seaweed Hydrocolloid and Gac Pulp (*Momordica cochinchinensis*) Edible Film

Thuy Thi Bich Tran [1,2,*], Boi Ngoc Vu [2], Md Saifullah [1], Minh Huu Nguyen [1,3], Penta Pristijono [1], Timothy Kirkman [1] and Quan Van Vuong [1,*]

[1] College of Engineering, Science and Environment, School of Environmental and Life Sciences, The University of Newcastle, 10 Chittaway Road, Ourimbah, NSW 2258, Australia; md.saifullah@uon.edu.au (M.S.); minh.nguyen@newcastle.edu.au (M.H.N.); penta.pristijono@newcastle.edu.au (P.P.); Timothy.Kirkman@newcastle.edu.au (T.K.)

[2] Faculty of Food Technology, Nha Trang University, Nha Trang City 058, Vietnam; boivn@ntu.edu.vn

[3] School of Science and Health, Western Sydney University, Penrith, NSW 2751, Australia

* Correspondence: thuyttb@ntu.edu.vn (T.T.B.T.); vanquan.vuong@newcastle.edu.au (Q.V.V.); Tel.:+61-4068-74263 (T.T.B.T.); +61-4101-55781 (Q.V.V.)

**Abstract:** Edible films and coatings have currently received increasing interest because of their potential in food applications. This study examined the effect of incorporated essential oils and natural plant extracts on the characteristics of the composite seaweed hydrocolloid and gac pulp films. Films were prepared by a casting technique, followed by measurement of physical, optical, barrier, mechanical, and structural properties. The results showed that adding plant oils and extracts significantly affected the physical, optical, mechanical, and structural properties of the composite films. Incorporation of the essential oils resulted in a reduction in moisture content and opacity while increasing values for Hue angle and elongation at break of the composite films. Besides, incorporation of the plant extracts showed increases in thickness, opacity, ΔE, Chroma, and elongation at the break, while there is a decrease in the Hue angle values of the composite films. In conclusion, incorporating plant essential oils and extracts into composite seaweed hydrocolloid and gac pulp films can enhance film properties, which can potentially be applied in food products.

**Keywords:** edible film; essential oils; plant extracts; characterization



## 1. Introduction

Edible films and coatings are considered as an alternative to tackle the negative impacts of synthetic packaging. Edible films and coatings are thin layers of edible materials applied as primary packaging for foods [1,2]. They provide barriers to oxygen, moisture, and solute movement to preserve the quality of food from deterioration [3]. Primary film-forming ingredients include polysaccharides [4–6], proteins [7,8], and lipids [9–11]. Plasticizers and additives are also added to improve their functional properties.

Polysaccharides from seaweed, such as sodium alginate and kappa-carrageenan, have been widely used as structural materials [12]. Sodium alginate is an antitumor, anticoagulant, and antiviral agent, whereas kappa-carrageenan is known as a thickener, stabilizer, gas barrier and food appearance enhancer [2,13]. However, the edible films and coatings produced from the single polysaccharides typically have poor water vapour barrier properties [14]. Therefore, composite films are preferred for extending the shelf life of fruits, vegetables, fish and meat products [2]. There are many compounds to be combined with polysaccharides to improve edible film properties, such as essential oils to increase water resistance [3], moisture barrier, and oxygen permeability of the films [15,16]. Besides, natural plant extracts, such as pomegranate peel extract, grape seed extract, green tea

extract, turmeric extract, macadamia skin and blueberry ash have been added to improve the edible film and coating properties [17–21].

Food by-products are becoming increasingly common sources of ingredients to develop edible films and coatings as either structural materials or as added bioactive compounds [22–25]. Recent studies on gac pulp (*Momordica cochinchinensis*), a waste product from gac oil production, have shown that it contains significant levels of bioactive compounds, especially carotenoids and phenolic compounds [26–28]. Our recent study also revealed that gac pulp could improve the physical and mechanical properties of the films [29]. As gac pulp is an inexpensive material, it has the potential to be incorporated into edible films and coatings to improve their properties and potentially reduce the cost.

This study aimed to explore the influence of different essential oils and plant extracts on physical, colour, barrier, and mechanical properties of seaweed hydrocolloid/gac pulp-based edible films.

## 2. Materials and Methods

### 2.1. Materials

Certified kappa-carrageenan powder (E407, Chondrus crispus extract, Philippines origin), certified sodium alginate powder (E410, Chile origin), and peppermint essential oil were supplied by the Melbourne Food Depot (Melbourne, Australia). Ginger and lemongrass essential oils were purchased from Sigma Andrich. Lemon myrtle essential oil was purchased from Essentially Australia (New South Wales, Australia). Food-grade glycerol was supplied by Ajax Finechem (New South Wales, Australia). Food-grade gac oil was purchased from VNPOFOOD Company, Vietnam.

Mature Gac fruit samples were selected from central markets in Khanh Hoa province, Vietnam. After cleaning, the pulp was separated, sliced, and freeze-dried using an industrial freeze-dryer model LyoBeta 35 (Telstar Technologies, Valencia, Spain) for 24 h. Lemon myrtle extract, blueberry ash fruit extract, and macadamia peel extract were prepared according to the methods described by Saifullah, McCullum, McCluskey and Vuong [30], Vuong, Ngoc Thuy Pham, Vu, Dang, Van Ngo and Chalmers [31] and Dailey and Vuong [32].

### 2.2. Preparation of Gac Pulp-Based Films with Plant Essential Oils and Extracts

Edible films were produced by a casting process. Control solutions were prepared as described in our previous study [33]. Briefly, seaweed hydrocolloids, including 1.03% (*w/v*) sodium alginate, 0.65% (*w/v*) kappa-carrageenan combined with 0.4% (*w/v*) gac pulp were dissolved in deionized water under control heating (65 °C) and continuous stirring. The pectin-based solution was 1.28% (*w/v*) sodium alginate, 0.58% *w/v* kappa-carrageenan, and 0.25% *w/v* gac pulp pectin that was extracted from gac pulp powder. The film-forming solution was cooled to 50 °C and glycerol (0.85% *w/v*) was then added as a plasticizer. Different plant essential oils, including peppermint, ginger, lemongrass, lemon myrtle, and gac oils and other natural plant extracts, such as lemon myrtle extract, blueberry ash, and macadamia extract, were then separately added to the solution mixtures. All suspension solutions were stirred for a further 5 min. There are ten different formulas (Table 1) prepared, including GP: gac pulp control, GPP: gac pulp pectin, GGG: gac pulp ginger, GLG: gac pulp lemongrass, GPM: gac pulp peppermint, GLO: gac pulp lemon myrtle oil, GGO: gac pulp gac oil, GLE: gac pulp lemon myrtle extract, GBA: gac pulp blueberry ash extract, GMC: gac pulp macadamia extract.

**Table 1.** Composition of the studied films incorporating essential oils and plant extracts.

| Materials % (*w/v*) | | Samples | 1 GP | 2 GPP | 3 GGG | 4 GLG | 5 GPM | 6 GLO | 7 GGO | 8 GLE | 9 GBA | 10 GMC |
|---|---|---|---|---|---|---|---|---|---|---|---|---|
| Film forming materials | Seaweed hydrocolloids | Sodium alginate | 1.03 | 1.28 | 1.03 | 1.03 | 1.03 | 1.03 | 1.03 | 1.03 | 1.03 | 1.03 |
| | | Kappa-carageenan | 0.65 | 0.58 | 0.65 | 0.65 | 0.65 | 0.65 | 0.65 | 0.65 | 0.65 | 0.65 |
| | Gac pulp | Gac pulp powder | 0.4 | - | 0.4 | 0.4 | 0.4 | 0.4 | 0.4 | 0.4 | 0.4 | 0.4 |
| | | Gac pulp pectin | - | 0.25 | - | - | - | - | - | - | - | - |
| Plasticizer | | Glycerol | 0.85 | 0.85 | 0.85 | 0.85 | 0.85 | 0.85 | 0.85 | 0.85 | 0.85 | 0.85 |
| Essential oils | | Ginger (*Zingiber officinale*) oil | - | - | 0.15 | - | - | - | - | - | - | - |
| | | Lemongrass (*Cymbopogon*) oil | - | - | - | 0.15 | - | - | - | - | - | - |
| | | Peppermint (*Mentha x piperita*) oil | - | - | - | - | 0.15 | - | - | - | - | - |
| | | Lemon myrtle (*Backhousia citriodora*) oil | - | - | - | - | - | 0.15 | - | - | - | - |
| | | Gac oil | - | - | - | - | - | - | 0.15 | - | - | - |
| Natural plant extracts | | Lemon myrtle | - | - | - | - | - | - | - | 0.15 | - | - |
| | | Blue berries (*Vaccinium sect. Cyanococcus*) ash | - | - | - | - | - | - | - | - | 0.15 | - |
| | | Macadamia | | | | | | | | | | 0.15 |

All films were formulated by casting 20 g of the suspension solution in a Petri dish (10 cm in diameter) and then dried in an oven at 30 °C for 24 h. Dried films were peeled off from the petri dishes and conditioned at 75% relative humidity (RH) and 30 °C for 72 h prior to testing [34].

*2.3. Determination of Edible Film Characteristics*

2.3.1. Physical Properties

Film Thickness

The thickness was determined using a digital micrometer Model ID-F125 (Mitutoyo, Co., Kanagawa, Japan) as described by Thakur et al. (2017) (*n* = 10). The instrument's precision was 0.001 mm. The film thickness was used for further calculation of the optical and barrier properties.

Moisture Content

The moisture content was measured using the thermos-gravimetric method as previously described by Saberi, Thakur, Vuong, Chockchaisawasdee, Golding, Scarlett and Stathopoulos [35] with some modifications. The film was cut into 40 mm × 15 mm specimens. The cut samples were initially weighted ($M_i$) and then dried at 105 °C in a hot air Labec oven (Laboratory Equipment, New South Wales, Australia) and reached a constant weight. The final weight ($M_f$) was recorded to calculate the weight loss of individual spec-

imens with three replicates. Moisture content of obtained films was measured according to the following Equation (1):

$$Moisture\ content\ (\%) = \frac{M_i - M_f}{M_i} \times 100 \tag{1}$$

Opacity

The opacity was determined using a UV–vis Spectrophotometer (Cary 50 Bio, Melbourne, Australia) as reported by Abdalrazeq, Giosafatto, Esposito, Fenderico, Di Pierro and Porta [36] with some modifications. Film strips (10 mm × 50 mm) were placed in a quartz cuvette to measure the light absorbance at 560 nm. Film samples were measured in triplicate. Calculation of opacity was based on the absorbance divided by the film thickness as the following equation:

$$Opacity = \frac{Abs_{560}}{T} \tag{2}$$

where $Abs_{560}$: the absorbance at 560 nm and $T$ was the thickness of the obtained film (mm).

Colour

The colour attributes were measured using a colorimeter (CR-300, Konica Minolta, Tokyo, Japan) according to a previous method [25]. The instrument was calibrated using a white colour plate as a background. Difference measurements of an individual film were performed by lightness (L), red-green (a), and yellow-blue (b). The total colour difference (ΔE) was directly provided by the instrument. Chroma and Hue angle, were established by Equations (3)–(5) respectively. Mean of ten different measurements was taken as the colour result of an individual film.

$$Chroma = \sqrt{a^2 + b^2} \tag{3}$$

$$\text{Hue angle = Degrees (arctan (b/a)) if a > 0} \tag{4}$$

$$\text{Hue angle} = 180° + \text{Degrees (arctan (b/a)) if a < 0} \tag{5}$$

2.3.2. Barrier Properties
Water Vapour Permeability

Water vapour permeability was determined following the method of Moghadam, Salami, Mohammadian, Khodadadi and Emam-Djomeh [25] based on an ASTM procedure [37]. Prior to measurements, all film samples were conditioned in a desiccator at 75% relative humidity (RH). Each pre-conditioned film was then placed over the opening of circular permeation test cup (0.7065 mm$^2$ film area) that contained 30 g anhydrous calcium chloride (RH 0%) and was tightly sealed by melted paraffin. After that, the film containing cups were placed in a desiccator with saturated NaCl aqueous solution (70% RH). The weight changes of the test cups were periodically documented every 2 h over 24 h. The changes were plotted as a function of time. The weight gain $\Delta m$ of the cell per time unit $\Delta t$ was used to calculate the water vapour permeability (WVP, g Pa$^{-1}$ s$^{-1}$ m$^{-1}$) according to Equation (6).

$$WVP = \frac{\Delta m}{A\ \Delta t} \times \frac{T}{\Delta P} \tag{6}$$

where,
$\frac{\Delta m}{\Delta t}$: was determined by the slope of the straight line (gs$^{-1}$)
$A$: Surface area (m$^2$)
$T$: thickness (mm)
$\Delta P$: the water vapour pressure difference inside and outside of the film (Pa)

2.3.3. Mechanical Properties

Elongation at break (EAB) and tensile strength (TS) were evaluated according to Tran, Vu, Pristijono, Kirkman, Nguyen and Vuong [38]. Firstly, the film samples were

equilibrated at relative humidity of 75% until constant weight. Then, rectangular pieces (15 mm × 40 mm) were cut and gripped by two jaws of a Food Texture analyser (LLOYD, London, UK). The maximum tensile (N/m) and elongation (mm) at break down point were measured at initial grip distance of 40 mm and speed of 1 mm/s. There were 5 replicates conducted for each film.

### 2.3.4. Structural Characterization
Scanning Electron Microscopy

Scanning electron microscopy (SEM) of the test edible films was determined as previously reported [39]. The film samples were placed on a double-sided tape with an aluminium specimen holder and the image was taken using a Zeiss VP scanning electron microscope (Sigma, Missouri, United States) at an accelerating voltage of 1.00 kV and an aperture size chosen at 30 μm.

X-ray Diffraction

X-ray diffraction (XRD) was applied to examine the difference in structure of obtained edible films using the X-ray diffractometer (PHILIPS). Each film specimen was scanned at 25 °C at a diffraction angle of 2θ ranging from 5° to 50°. The certain step-size was at 0.02 per second.

### *2.4. Statistical Analysis*

Statistical analysis was implemented using the JMP statistical software, version Pro 14 (SAS institute, North Carolina, USA). One-way analysis of variance (ANOVA) was applied to compare the significant differences between different formulas of edible film. The statistical significance was established at $p < 0.05$.

## 3. Results and Discussion
### *3.1. Impact of Essential Oils and Plant Extracts on Thickness, Moisture Content, and Opacity of the Seaweed Hydrocolloid and Gac Pulp Edible Films*
#### 3.1.1. Film Thickness

Thickness is an important parameter of the film and can be affected by the film-forming material and its composition [40]. In this study, thickness of the seaweed hydrocolloid/gac pulp edible films ranged from around 0.06 mm to 0.10 mm and was significantly ($p < 0.05$) affected by the inclusion of essential oils and plant extracts (Figure 1). GPP films had the lowest thickness, with of 0.06 mm. Overall, the GPP-based films were thinner than other films. These findings are in alignment with a previous study by Galus and Lenart [41] who reported that the thickness of pectin-based edible films was lower than that of other polysaccharide-based edible films. The results also showed that the thickness of edible films which incorporated gac oil, blueberry ash and macadamia extract was approximately 0.1 mm, which was thicker than the films without additives, which had a thickness of less than 0.08 mm. These findings are in agreement with a previous study by Nazurah and Hanani [42] who reported that edible films incorporated with plant extracts were thicker than the films without additives. In addition, the results also indicated that the type of essential oils and plant extracts affected the thickness of the films. The films incorporated with gac oil were thicker than the films incorporated with other essential oils. In the literature, film thickness has been reported to depend not only on processing parameters, but also on film composition [43]. Therefore, the nature of solid material of particular additives may effect the way they enter and distribute in the film network differently [44]. As a result, film which incorporated gac oil, blueberry ash extracts and macadamia extract were thicker than films prepared without additives, while all other additives showed no significant effect on the thickness of the film.

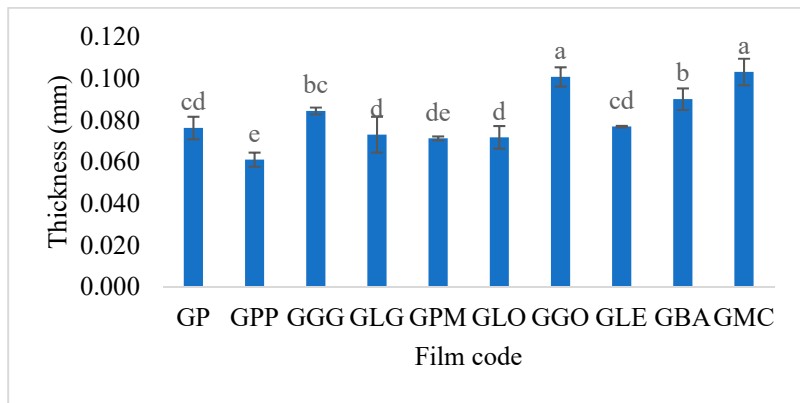

**Figure 1.** Thickness of studied edible films incorporating essential oils and plant extracts. Data are means ± standard deviations. Data of each column not sharing similar letters are significantly different, at $p < 0.05$.

### 3.1.2. Moisture Content

Moisture content of the studied edible films was presented in Figure 2. The results showed that gac pulp-based films (30.87%) and the films incorporated with macadamia extract (30.89%) had a higher moisture content than that of other films ($p < 0.05$). Overall, the films incorporated with essential oils had lower moisture contents. The result was supported by previous studies, which have indicated that incorporation of oil into the films could lead to a lower moisture content in the films [42,45]. The moisture content reduction was explained by Bourbon, Pinheiro, Cerqueira, Rocha, Avides, Quintas and Vicente [46] due to the hydrophobicity of oils. Of note, incorporation of plant extracts seems to have had various impacts on the moisture content of the films. There was no significant difference between the moisture content of gac pulp-based film and those incorporated with lemon myrtle, blueberries ash and macadamia extracts. However, the film made from gac pulp pectin had a lower moisture content than that of the gac pulp film, specifically 26.08% compared to 30.87%. The lower moisture content with the pectin could be explained by the lesser thickness of the films [47].

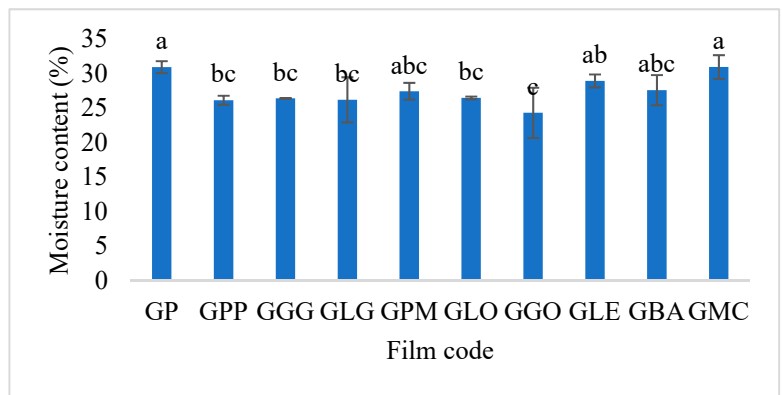

**Figure 2.** Moisture content of studied edible films incorporating essential oils and plant extracts. Data are means ± standard deviations. Data of each column not sharing similar letters are significantly different, at $p < 0.05$.

### 3.1.3. Opacity

Opacity measures the impenetrability of the film to visible light, with both opaque (high opacity) and transparent (low opacity) films having utility in food industry applications [48]. The results (Figure 3) indicated that the gac pulp pectin edible film had the lowest opacity, at around 0.97%. The clear and transparent films based on pectin and



alginate were previously reported by Galus and Lenart [41]. The results also revealed that the films incorporated with essential oils also had lower opacity compared to the control gac pulp film, except for the GLG film, the opacity of which was at around 3.5%. Our results are in agreement with a previous study that found that kappa-carrageenan films which incorporated different plant oils had a lower opacity [42]. In contrast, the addition of plant extracts, including lemon myrtle, blue berries ash, and macadamia extract, increased the film opacity (Figure 3). Opacity of GLE was 1.5 times higher than that of the gac pulp film with no additives. The increase in opacity showed a lesser transparency in edible films mixed with plant extracts. A similar result was also reported by Wang, Dong, Men, Tong and Zhou [49], that the addition of plant extracts caused a higher degree of opacity. This was explained by Chana-Thaworn, Chanthachum and Wittaya [50] because of the delay in light transmission caused by the incorporation of plant extracts.

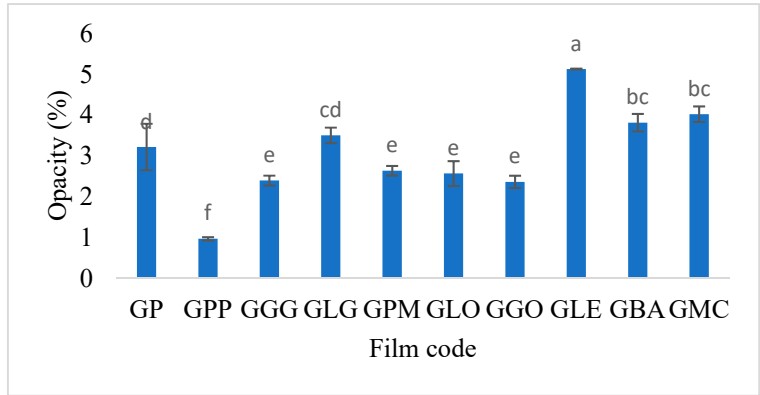

**Figure 3.** Opacity of studied edible films incorporating essential oils and plant extracts. Data are means ± standard deviations. Data of each column not sharing similar letters are significantly different at *p* < 0.05.

3.1.4. Colour of the Films

The colour of a film plays a vital role in its application to food products [51]. This is because colour properties can affect the consumer acceptability [47]. The colour of the films is typically reflected by lightness (L), redness (a), greenness (b), total colour difference (ΔE), Chroma and Hue angle [52]. The results (Table 2) showed that GP film had similar colour to the GPP in term of L values (around 95.55) and Hue angle (around 94). These findings were similar to the results from a previous study by Tran, Roach, Nguyen, Pristijono and Vuong [33]. However, there was a significant difference in colour between films which incorporated essential oils and plant extracts. Of note, there was no significant difference in the lightness of GP film compared with essential oil added films, such as GGG, GLG, GPM, GLO, and GGO (Table 2). The results contradicted previous studies, where the addition of essential oils, depending on their nature, could either increase [46] or decrease [48,53] in the lightness of the films.

The results showed that the GP film and the films incorporated with essential oils had negative values for greenness and were positive for redness, whereas the films incorporated with plant extracts had positive values for both greenness and redness (Table 2). It is interesting to note that the film incorporated with gac oil had a significant difference in colour attributes in comparison with other films. This is due to the high levels of carotenoids found in gac oil, which produce an intense red colour [54]. The results further indicated that films incorporating essential oils had higher Hue angles, ranging from 95 to 98, as compared to that of the gac pulp control film (94.05 ± 0.3). Of note, the film incorporated with gac oil had the highest value of Hue angle (97.54 ± 0.2). The results also showed that the films incorporating plant extracts generally had lower values of Hue angle, ranging from 82.77 ± 0.23 to 84.6 ± 0.25 (Table 2). This is due to the higher concentration of polymeric colour in plant extract-based films [55].

**Table 2.** Effects of essential oils and plant extracts on colour of the seaweed hydrocolloid and gac pulp edible films.

| Samples | L | a | b | ΔE | Chroma | Hue Angle |
|---------|---|---|---|-----|--------|-----------|
| GP | 95.55 ± 0.33 [a] | −0.33 ± 0.03 [d] | 4.63 ± 0.09 [ef] | 2.2 ± 0.90 [efg] | 4.95 ± 0.99 [ef] | 94.05 ± 0.3 [e] |
| GPP | 95.55 ± 0.44 [a] | −0.34 ± 0.05 [d] | 4.21 ± 0.32 [f] | 1.87 ± 0.39 [g] | 4.22 ± 0.32 [f] | 94.55 ± 0.38 [de] |
| GGG | 95.82 ± 0.06 [a] | −0.41 ± 0.02 [de] | 4.61 ± 0.13 [ef] | 2.09 ± 0.14 [fg] | 4.63 ± 0.13 [ef] | 95.13 ± 0.17 [cd] |
| GLG | 95.86 ± 0.08 [a] | −0.55 ± 0.03 [e] | 5.34 ± 0.13 [d] | 2.78 ± 0.15 [de] | 5.36 ± 0.13 [d] | 95.88 ± 0.2 [b] |
| GPM | 95.77 ± 0.05 [a] | −0.42 ± 0.01 [de] | 4.57 ± 0.16 [ef] | 2.07 ± 0.15 [fg] | 4.59 ± 0.16 [ef] | 95.2 ±0.15 [cd] |
| GLO | 95.7 ± 0.06 [a] | −0.56 ± 0.01 [e] | 5.19 ± 0.08 [de] | 2.69 ± 0.09 [def] | 5.22 ± 0.08 [de] | 96.21 ± 0.1 [b] |
| GGO | 95.8 ± 0.16 [a] | −0.76 ± 0.03 [f] | 5.78 ± 5.24 [d] | 3.26 ± 0.27 [d] | 5.83 ± 0.24 [d] | 97.54 ± 0.2 [a] |
| GLE | 79.87 ± 0.24 [d] | 2.27 ± 0.18 [a] | 23.95 ± 0.86 [a] | 27.11 ± 0.63 [a] | 24.06 ± 0.87 [a] | 84.6 ± 0.25 [f] |
| GBA | 86.55 ± 0.31 [b] | 1.27 ± 0.04 [c] | 11.83 ± 0.31 [c] | 13.6 ± 0.29 [c] | 11.9 ± 0.3 [c] | 83.87 ± 0.53 [g] |
| GMC | 83.55 ± 0.33 [c] | 2.03 ± 0.08 [b] | 16.02 ± 0.16 [b] | 18.72 ± 0.29 [b] | 16.15 ± 0.16 [b] | 82.77 ± 0.23 [h] |

Data are means ± standard deviations. Data in the same column not sharing similar letters are significantly different at $p < 0.05$.

### 3.2. Impact of Essential Oils and Plant Extracts on Barrier and Mechanical Properties of the Seaweed Hydrocolloid and Gac Pulp Edible Films

#### 3.2.1. Water Vapour Permeability

Water vapour permeability (WVP) reflects the capacity of moisture transmission, and therefore films' ability to prevent water loss during the preservation of coated products. The results (Table 3) showed that the GP film had a significantly higher WVP ($1.76 \pm 0.008$ g Pa$^{-1}$ s$^{-1}$ m$^{-1}$) than that of the GPP edible film ($1.25 \pm 0.011$ g Pa$^{-1}$ s$^{-1}$ m$^{-1}$) and GLO ($1.34 \pm 0.015$ g Pa$^{-1}$ s$^{-1}$ m$^{-1}$). However, WVP of the GP film was not significantly different to that of the films incorporated with other essential oils or plant extracts. The range of WVP was similar to that in a previous study by Tran, Roach, Nguyen, Pristijono and Vuong [33].

**Table 3.** Effects of essential oils and plant extracts on water vapour permeability and mechanical properties of the studied edible films.

| Samples | WVP (×10$^{-10}$ g Pa$^{-1}$ s$^{-1}$ m$^{-1}$) | EAB (mm) | TS (N/m) |
|---------|-----------------------------------------------|----------|----------|
| GP | 1.76 ± 0.08 [ab] | 15.76 ± 0.26 [c] | 1364.78 ± 92.10 [ab] |
| GPP | 1.25 ± 0.011 [d] | 19.76 ± 1.67 [ab] | 1372.27 ± 72.72 [ab] |
| GGG | 1.49 ± 0.08 [bcd] | 19.17 ± 0.18 [b] | 1376.46 ± 243.26 [ab] |
| GLG | 1.68 ± 0.004 [ab] | 15.69 ± 0.82 [c] | 1514.59 ± 263 [a] |
| GPM | 1.81 ± 0.006 [ab] | 18.10 ± 0.95 [b] | 1380 ± 147.53 [ab] |
| GLO | 1.34 ± 0.015 [cd] | 18.74 ± 1.30 [b] | 1069.81 ± 180.65 [cd] |
| GGO | 1.6 ± 0.051 [bc] | 21.99 ± 0.49 [a] | 1180.26 ± 97 [bc] |
| GLE | 1.62 ± 0.32 [bc] | 19.30 ± 0.54 [b] | 878.41 ± 187.81 [d] |
| GBA | 1.67 ± 0.31 [ab] | 18.76 ± 0.7 [b] | 1235.41 ± 212.19 [abc] |
| GMC | 1.98 ± 0.089 [a] | 17.62 ± 1.7 [bc] | 1377.48 ± 89.34 [ab] |

Data are means ± standard deviations. Data in the same column not sharing similar letters are significantly different at $p < 0.05$.

#### 3.2.2. Mechanical Properties

Mechanical properties, including elongation at break (EAB) and tensile strength (TS) of the films, were measured and the results are showed in Table 3. GLG and GMC films showed no significant difference in EAB compared to the control film. However, all other tested additives produced an increase to EAB, with gac oil producing the largest increase ($21.99 \pm 0.49$ mm). The results indicated that incorporation of essential oils and plant extracts tend to increase the EAB of the films. The results are supported by previous studies [42,48,56–58], and can be explained by the formation of cross-links that contribute to better cohesive and flexible structure of the composite films when adding essential oils.

The results also revealed that addition of lemon myrtle oil and extract significantly reduced the tensile strength of gac pulp-based edible films ($p < 0.05$). There was no significant difference in tensile strength caused by the inclusion of any other tested ad-

ditives. This is in agreement with previous studies [42,57,59,60]. The decrease in TS can be explained by the weak bonds between lemon myrtle oil and extract with film-forming polymers [17,59]. Thus, the current findings revealed that incorporation of plant oils and extracts can significantly increase EAB, but may also reduce the TS of the films.

### 3.3. Impact of Essential Oils and Plant Extracts on Structure of the Seaweed Hydrocolloid and Gac Pulp Edible Films

3.3.1. Surface and Cross-Section Microstructure

Scanning electron micrographs of film surfaces and cross-sections are shown in Figure 4. The results suggest that the GPP film had the smoothest and most homogeneous surface and cross-section microstructure. The smooth and homogeneous surface and cross-section microstructure can be explained by a well-ordered matrix between alginate, carrageenan and gac pulp pectin [39]. Overall, micrographs of GP films and other incorporated films also suggest a heterogeneous microstructure that correlates with the lower transparency of these films. The results also indicate that there were more convex irregularities on the surface of the oil-added films than the GP film. A previous study reported by [61] observed that incorporation of the essential oils and plant extracts could result in variation of structures due to the arrangement of film-forming polysaccharides and aggregation of oils and extracts during film production. It should be noted that the films incorporated with essential oils had more homogenous surfaces than that of the films incorporated with the plant extracts (Figure 4). In addition, the results also revealed that all the films had a fibrous-like structure, which concurs with previous studies on semi-refined kappa-carrageenan edible films [62] and alginate/ginseng films [59].

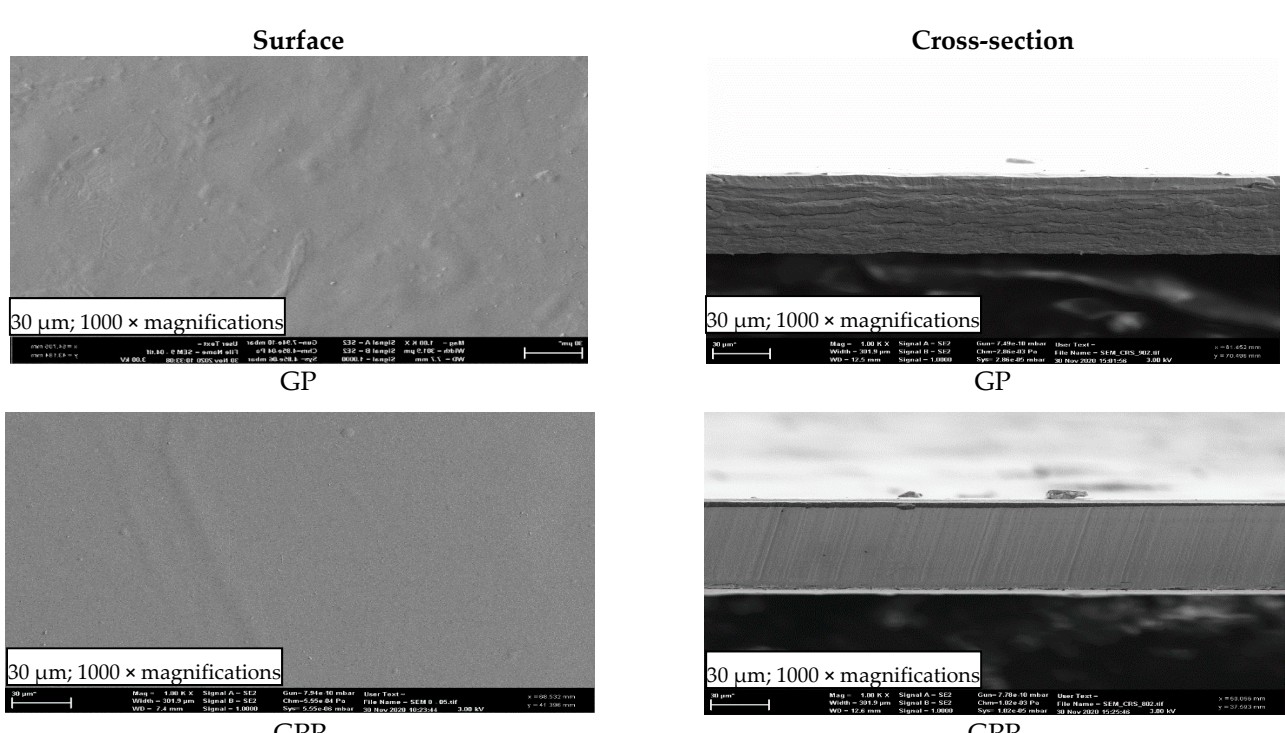

**Figure 4.** *Cont.*

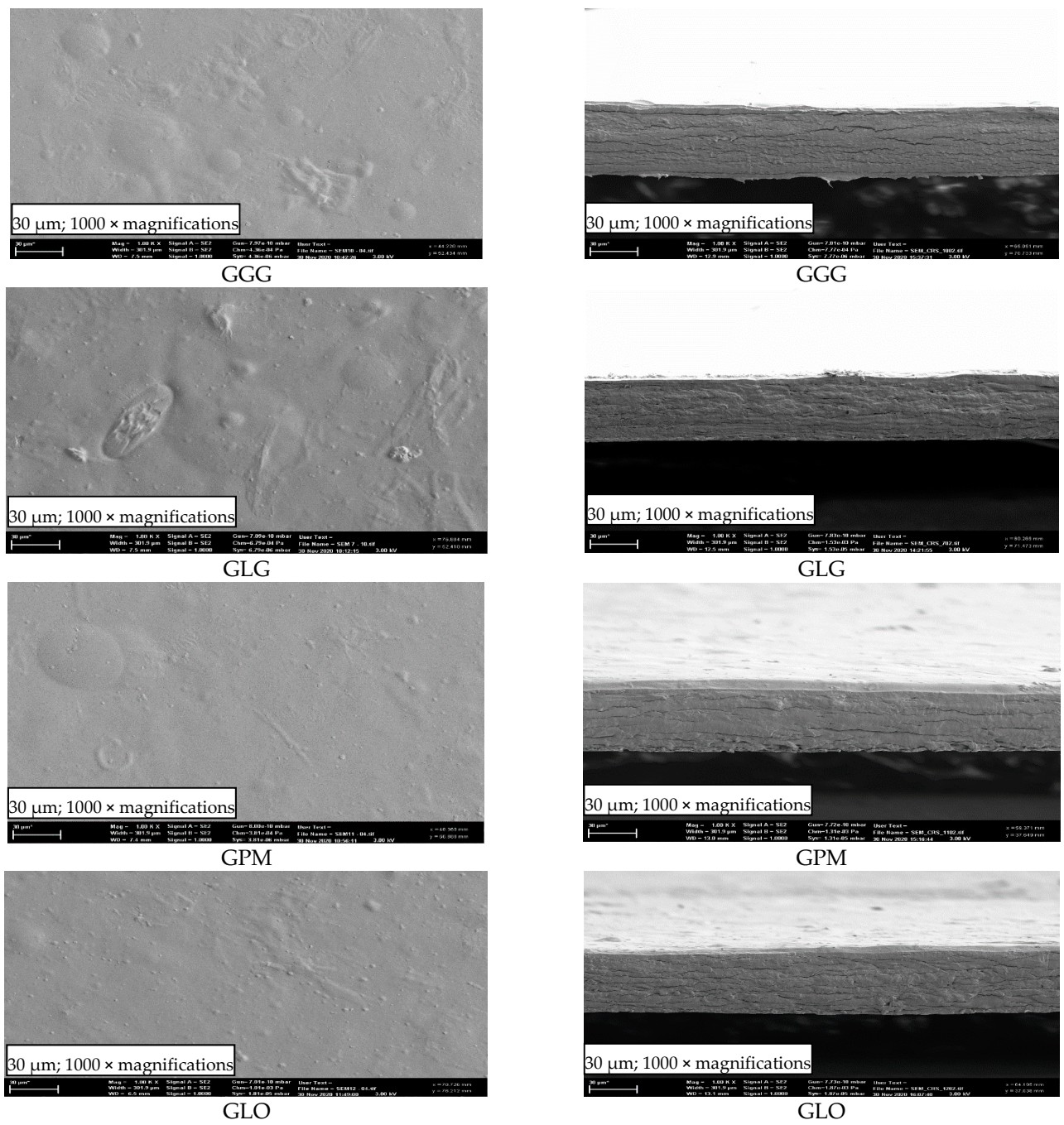

**Figure 4.** *Cont.*

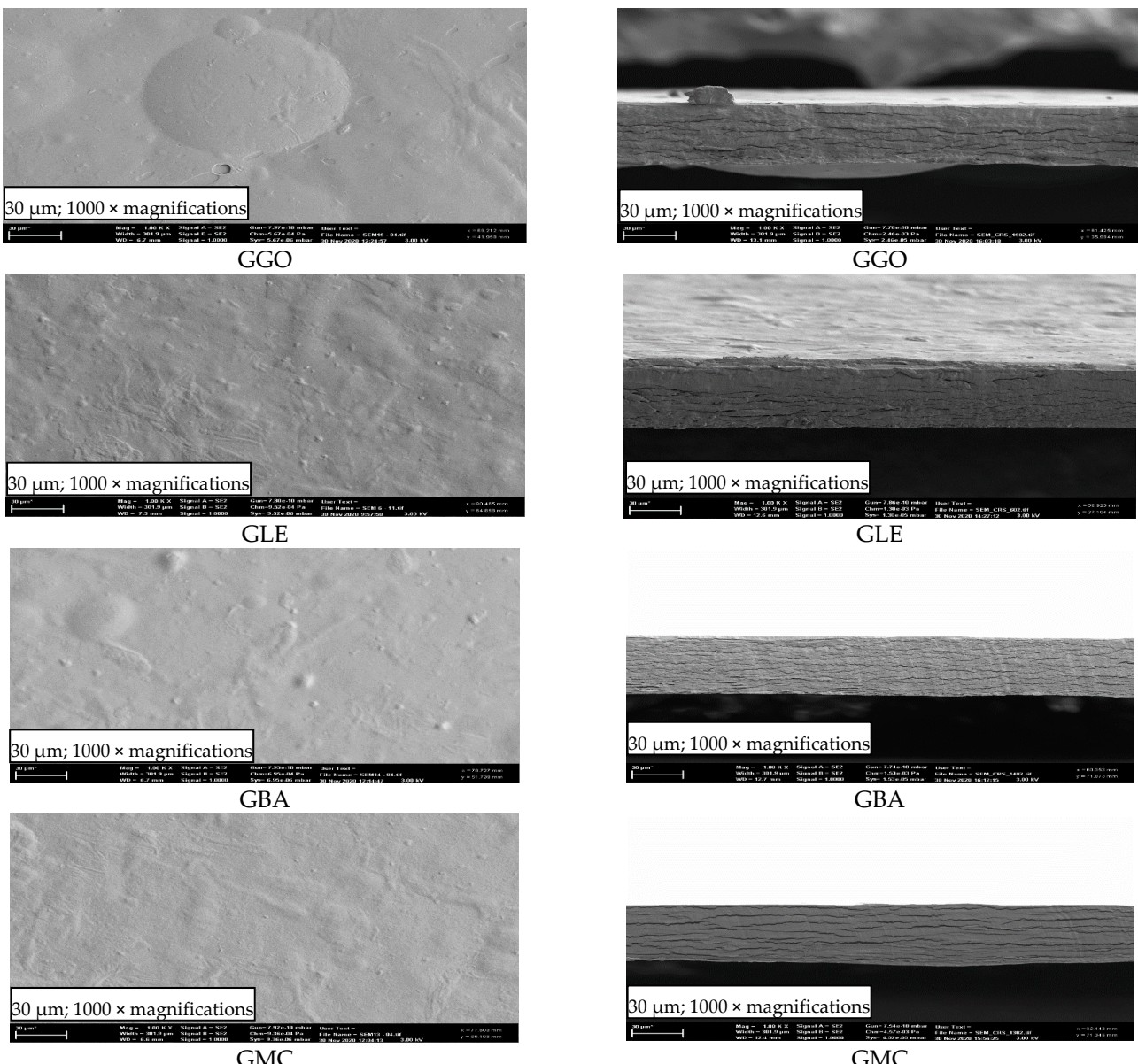

**Figure 4.** Scanning electron micrographs of the surface and cross-section of the gac pulp-based films.

### 3.3.2. Crystalline Structure

The crystalline structure of the films was measured using X-ray diffraction (XRD) analysis, and these results are showed in Figure 5.

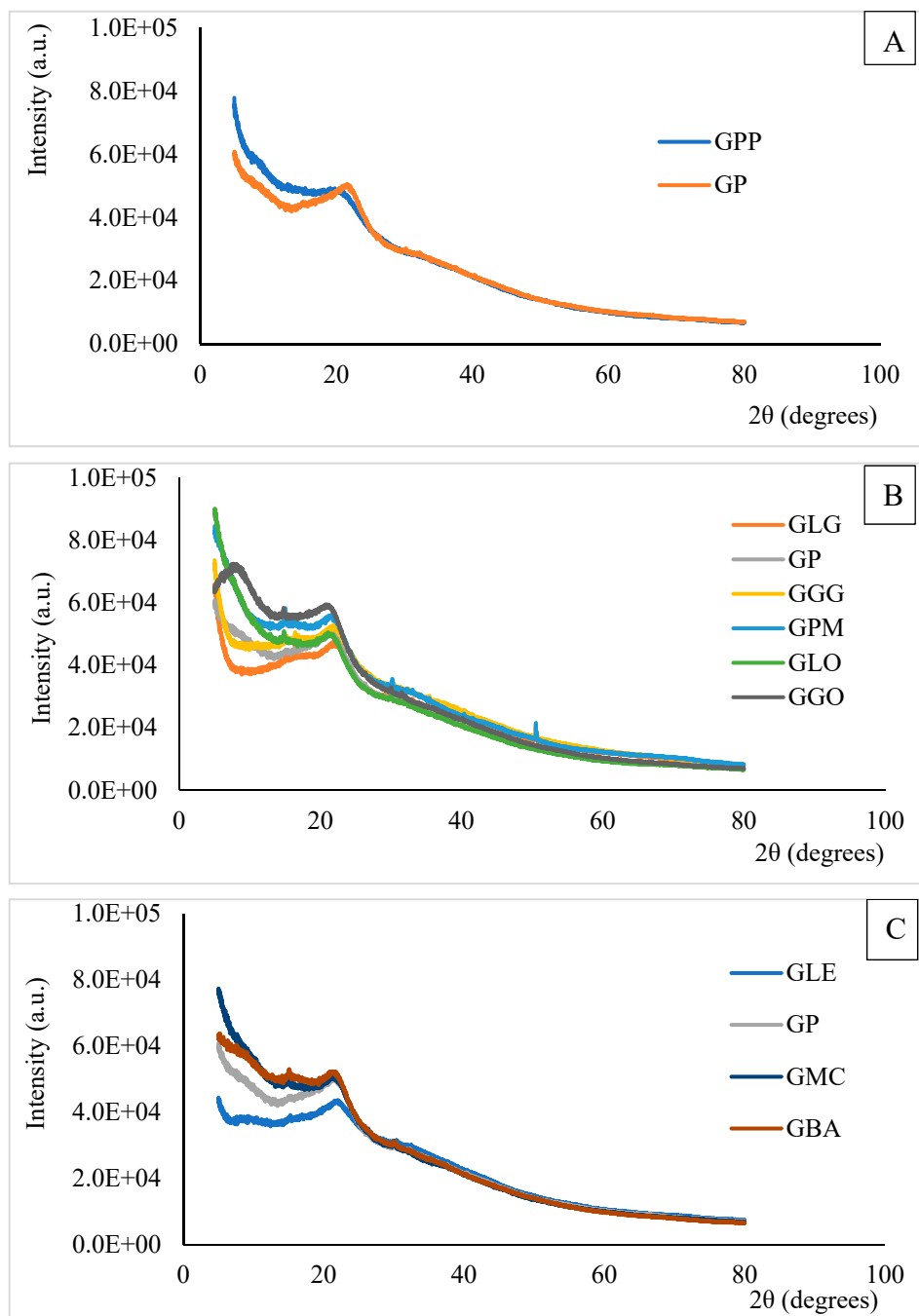

**Figure 5.** XRD patterns of control GP film and GPP film (**A**); of control GP film and films incorporated with plant oils (**B**); of control GP film and films incorporated with plant extracts (**C**).

XRD patterns of the control GP film and the GPP film were presented in Figure 5A, with both exhibiting the diffraction with a single peak. However, GP film showed a shaper peak as compared with that of GPP film, especially at 21.6° (2θ). The results indicate that the GP edible film had a semi-crystalline structure, whereas the GPP had an amorphous form. These findings can be explained by the diffraction of the films due to pectin, which was predominately amorphous in nature [63].

The diffraction of the films incorporated with the essential oils and plant extracts are showed in Figure 5B,C respectively. Overall, all the films had similar patterns of diffraction with a diffraction peak at around 21° (2θ). The results (Figure 5B) suggest that the films incorporated with the essential oils had more discernible peaks than the control gac pulp

film, indicating that addition of essential oils improve the crystalline structure of the films. These can be explained by the molecular interactions between the film components. The result was in line with previous studies [64,65].

The results (Figure 5C) also indicated that the films incorporated with the plant extracts had a relatively flat single peak (except the GMC film). Among them, GLE film showed a lower and less discernible peak at 21.11° (2θ) than the control gac pulp film, as well as GBA and GMC films. The results are supported by a study of Kanmani and Rhim [17]. Besides, the GMC film had another diffraction peak at 14.9°, revealing that incorporation of macadamia could affect the degree of crystalline in structure of edible film, while it was not in case of blueberry ash and lemon myrtle extracts.

## 4. Conclusions

Incorporation of plant oils and extracts significantly affected thickness, moisture content, opacity, colour, tensile strength and elongation at break of the studied edible films. It was found that the gac pulp pectin edible film had lower thickness, moisture content, opacity, and water vapour permeability, but a similar tensile strength and colour properties as the seaweed hydrocolloid and gac pulp film. Incorporation of plant oils tentatively decreased moisture content, opacity but increased values of Hue angle and elongation at the break of the films. In contrast, incorporation of plant extracts increased thickness, opacity, $\Delta E$, Chroma and elongation at break, but decreased Hue angle values as compared to the control GP film. The GPP film was found to have the smoothest surface. Addition of plant oils resulted in more crystalline structures than that of the control GP film. Overall, this study suggested that an incorporation of plant oils and extracts into seaweed hydrocolloid and gac pulp edible films, has great potential for tailoring the properties of these films for future applications in food.

**Author Contributions:** Conceptualization, M.H.N. and Q.V.V.; Data curation, T.T.B.T., M.S., P.P. and Q.V.V.; Formal analysis, T.T.B.T.; Funding acquisition, T.T.B.T.; Investigation, B.N.V. and M.H.N.; Methodology, T.T.B.T., B.N.V., M.S., P.P., T.K. and Q.V.V.; Project administration, Q.V.V.; Resources, T.T.B.T., B.N.V., M.H.N. and T.K.; Supervision, M.H.N., P.P., T.K. and Q.V.V.; Writing—original draft, T.T.B.T.; Writing—review and editing, T.T.B.T., B.N.V., M.S., M.H.N., P.P., T.K. and Q.V.V. All authors have read and agreed to the published version of the manuscript.

**Funding:** This research was funded by Vietnamese Government through the Ministry of Education and Training, Vietnam and The University of Newcastle, Australia, grant number 3141/QĐ-BGDĐT.

**Institutional Review Board Statement:** Not applicable.

**Informed Consent Statement:** Not applicable.

**Data Availability Statement:** The data supporting the research findings of this study are available from the corresponding author on request.

**Acknowledgments:** This work was supported by the Vietnamese Government through the Ministry of Education and Training, Vietnam and The University of Newcastle, Australia.

**Conflicts of Interest:** The authors declare no conflict of interest.

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
