# Peer review of "Impact of Various Essential Oils and Plant Extracts on the Characterization of the Composite Seaweed Hydrocolloid and Gac Pulp (Momordica cochinchinensis) Edible Film"

_processes, doi:10.3390/pr9112038_

Round 1
Reviewer 1 Report
- Concerning the manuscript “Impact of various essential oils and plant extracts on the characterization of 1 the composite seaweed hydrocolloid and Gac pulp edible film”
- There is no doubt that it summarizes valuable information concerning the topic of the study.
- Nevertheless, I am a little bit confused concerning the structure given to the methodology (2) since I expected they were planned to be given in terms of
2.3.1. Physical properties,
2.3.2. Barrier properties, and
2.3.3. Mechanical properties,
but I found in 69 physical, Colour parameters and optical properties, barrier, and mechanical properties and structural characterization …, Authors should consider colour and structural characterization as they are physical properties.
Secondly, I think the statistical analysis is common for all three properties It should not be included as a 2.4 as it is
- In 109 The film thickness is X and it would be convenient to be mentioned there, in spite it is mentioned for calculation of the optical and barrier properties. Besides it, in 157 thickness (mm) is T. are they the same?
- 141 Hue angle = 180o + Degrees (arctan (b/a)) if a > 0. Check the sign
- 252 Fig. 3. The opacity of studied edible films incorporating and 340 Fig. 4. Scanning electron micrographs. Check the different fonts
- Can the ordinate axis numbers in figure 5 be expressed with scientific notation?
- The information in the SEM images of figure 4 is not visible. I do not know if there is a need to show them all or only the most representative.
- Finally, I find both the titles of tables and the descriptions of section 3, Results and discussion, too long and repetitive.
For all the above I consider this manuscript accepted after the minor revision suggested.
Author Response
Dear Reviewer
We would like to thank the Editor and Reviewer for carefully reviewing our manuscript and giving very helpful comments for improvement.
Comments and Suggestions for Authors
- Concerning the manuscript “Impact of various essential oils and plant extracts on the characterization of the composite seaweed hydrocolloid and Gac pulp edible film”
- There is no doubt that it summarizes valuable information concerning the topic of the study.
- Nevertheless, I am a little bit confused concerning the structure given to the methodology (2) since I expected they were planned to be given in terms of
2.3.1. Physical properties,
2.3.2. Barrier properties, and
2.3.3. Mechanical properties,
but I found in 69 physical, Colour parameters and optical properties, barrier, and mechanical properties and structural characterization …, Authors should consider colour and structural characterization as they are physical properties.
Response: We would like to thank the Reviewer for the constructive comments. We agree with the Reviewer that structural characterization is a part of physical properties. Due to this is a large section with lots of information, we separated into two major sections to better show the results of the current study. We have revised the title of section 3.1 to avoid the confusion. It now reads “3.1. Impact of essential oils and plant extracts on thickness, moisture content, and opacity of the seaweed hydrocolloid and gac pulp edible films” instead of “Impact of essential oils and plant extracts on physical properties of the seaweed hydrocolloid and gac pulp edible films”
Secondly, I think the statistical analysis is common for all three properties It should not be included as a 2.4 as it is
Response: Although one-way analysis of variance (ANOVA) is a common for mean comparison, we believe that it is necessary to describe how the data were statistically analysed, especially the statistical significance value because this is linked with conclusions in the result section. Therefore, we remain the section 2.4.
- In 109 The film thickness is X and it would be convenient to be mentioned there, in spite it is mentioned for calculation of the optical and barrier properties. Besides it, in 157 thickness (mm) is T. are they the same?
Response: Thanks to the reviewer, we have corrected by using "T" as thickness in all formulations to avoid any confusion.
- 141 Hue angle = 180o + Degrees (arctan (b/a)) if a > 0. Check the sign
Response: The equation has been checked and corrected.
- 252 Fig. 3. The opacity of studied edible films incorporating and 340 Fig. 4. Scanning electron micrographs. Check the different fonts
Response: All tables and figures have been checked and corrected.
- Can the ordinate axis numbers in figure 5 be expressed with scientific notation?
Response: We keep the ordinate axis as taken from the X-ray diffractometer to maintain the original values of the image. We believe that the values do not affect the data interpretation and discussion, thus we would like to keep as it is.
- The information in the SEM images of figure 4 is not visible. I do not know if there is a need to show them all or only the most representative.
Response: The scale and magnification were presented in the figure description. We can provide the images with better resolution when the manuscript is accepted to enhance better visuality. We believe that it is better to show them all for comparison between essential oils and plant extracts.
- Finally, I find both the titles of tables and the descriptions of section 3, Results and discussion, too long and repetitive.
Response: We have used abbreviations for the titles and also checked the results and discussion to avoid repetition.
For all the above I consider this manuscript accepted after the minor revision suggested.
Response: Again, we would like to thank the Reviewer for useful comments.

Reviewer 2 Report
Dear Author,
The research released in article is very useful for farmers and environmentalists (as a method of using vegetable waste) and entrepreneurs in food industry as an alternative the negative impacts of synthetic packaging. The article has an interesting view of data, useful for farmers, entrepreneurs in food industry to better understand the edible films and coatings in different aspects (consumers, producers, farmers and environment).
The paper is well described and the methods used scientifically appropriate. There are few points should be addressed by author:
- The authors must use the Microsoft Word template or LaTeX template to prepare their manuscript. Using the template file will substantially shorten the time to complete copy-editing and publication of accepted manuscripts - the format of the text in the submitted paper is not the REQUIRED one.
- Please kindly verify and follow: https://www.mdpi.com/journal/processes/instructions#preparation for all your paper
- In the text, reference numbers should be placed in square brackets [ ], and placed before the punctuation; for example [1], [1–3] or [1,3]. For embedded citations in the text with pagination, use both parentheses and brackets to indicate the reference number and page numbers; for example [5] (p. 10). or [6] (pp. 101–105)
- It is useful to specify the ”latin” name of the ”gac” – the used raw material – in the title of the paper (as in the authors previously articles) and also mentioned in the paper
- In my opinion, it is important to specify the latin name of the plants used for oil extracts in Table 1.
- Please, pay attention the Table 1 should be in all in one page
- ”Sodium alginate” and ”Kappa-carageenan” should be mentioned in Table 1 that are seaweed products – for an easy understanding for the random readers
- Please verify the measurement unit in the Figure 1 – is ”mm” or other unit? Is not clear from the data presented in the paragraph – lines 192-194 and the data presented in Figure 1....
- In section - 3.1.2. Moisture Content – please specify at least some of your data presented in Figure 2! It is not enough only the following sentence ”Moisture content of studied edible films was presented in Fig. 2.” to someone clearly understand the data exposed
- In Figure 2, please specify the unit measure for opacity as ”%”.
- Line 238-239: Please complete the sentence ”The results (Fig. 3) indicated that the gac pulp pectin edible film had the lowest opacity” with the value exposed in the Fig.3.
- Line 239-240: Please complete the phrase: ”The results also revealed that the films incorporated with 240 essential oils also had lower opacity, except for the GPM film” with the value exposed in Fig 3.
- Line 260-261: Please complete the sentence ”The results (Table 2) showed that GP film had similar color to the GPP” with the values exposed in Table 2.
- The section ”Results and discussions” should be reorganize as two distinct sections ”Results” and ”Discussion” as demanded in instructions preparation
- All figures and tables names should be one the same page as figure or table, respectively
- Please insert the ”Author Contributions”
- Please also verify: https://www.mdpi.com/journal/processes/instructions#preparation for reference section
Thank you!

Author Response
Dear Reviewer
We would like to thank the Editor and Reviewer for carefully reviewing our manuscript and giving very helpful comments for improvement.
Dear Author,
The research released in article is very useful for farmers and environmentalists (as a method of using vegetable waste) and entrepreneurs in food industry as an alternative the negative impacts of synthetic packaging. The article has an interesting view of data, useful for farmers, entrepreneurs in food industry to better understand the edible films and coatings in different aspects (consumers, producers, farmers and environment).
The paper is well described and the methods used scientifically appropriate. There are few points should be addressed by author:
- The authors must use the Microsoft Word template or LaTeX template to prepare their manuscript. Using the template file will substantially shorten the time to complete copy-editing and publication of accepted manuscripts - the format of the text in the submitted paper is not the REQUIRED one.
- Please kindly verify and follow: https://www.mdpi.com/journal/processes/instructions#preparation for all your paper
- In the text, reference numbers should be placed in square brackets [ ], and placed before the punctuation; for example [1], [1–3] or [1,3]. For embedded citations in the text with pagination, use both parentheses and brackets to indicate the reference number and page numbers; for example [5] (p. 10). or [6] (pp. 101–105)
- It is useful to specify the ”latin” name of the ”gac” – the used raw material – in the title of the paper (as in the authors previously articles) and also mentioned in the paper
- In my opinion, it is important to specify the latin name of the plants used for oil extracts in Table 1.
- Please, pay attention the Table 1 should be in all in one page
Response: We would like to thank the Reviewer for the constructive comments. We have revised our manuscript based on the Reviewer’s suggestions.
- ”Sodium alginate” and ”Kappa-carageenan” should be mentioned in Table 1 that are seaweed products – for an easy understanding for the random readers
Response: We have revised as suggested by the Reviewer.
- Please verify the measurement unit in the Figure 1 – is ”mm” or other unit? Is not clear from the data presented in the paragraph – lines 192-194 and the data presented in Figure 1....
Response: Yes, it is "mm". The correction has been done as suggested by the Reviewer.
- In section - 3.1.2. Moisture Content – please specify at least some of your data presented in Figure 2! It is not enough only the following sentence ”Moisture content of studied edible films was presented in Fig. 2.” to someone clearly understand the data exposed
Response: The correction has been done as suggested by the Reviewer.
- In Figure 2, please specify the unit measure for opacity as ”%”.
Response: The correction has been done as suggested by the Reviewer.
- Line 238-239: Please complete the sentence ”The results (Fig. 3) indicated that the gac pulp pectin edible film had the lowest opacity” with the value exposed in the Fig.3.
Response: The correction has been done as suggested by the Reviewer.
- Line 239-240: Please complete the phrase: ”The results also revealed that the films incorporated with 240 essential oils also had lower opacity, except for the GPM film” with the value exposed in Fig 3.
Response: The correction has been done as suggested by the Reviewer.
- Line 260-261: Please complete the sentence ”The results (Table 2) showed that GP film had similar color to the GPP” with the values exposed in Table 2.
Response: The correction has been done as suggested by the Reviewer.
- The section ”Results and discussions” should be reorganize as two distinct sections ”Results” and ”Discussion” as demanded in instructions preparation
Response: As the “Instructions for Authors” of Process state that “the discussion can be combined with the Results”. We would like to remain them in combined format to show and discuss our findings.
- All figures and tables names should be one the same page as figure or table, respectively
- Please insert the ”Author Contributions”
- Please also verify: https://www.mdpi.com/journal/processes/instructions#preparation for reference section
Response: The correction has been done as suggested by the Reviewer. Thank you!

Reviewer 3 Report
The mnuscript have several points must be consiwdred previous to be cosnidered for publication in Processes.
- The references throw the text must be formatted according the mpdi guidelines,I mean the references must be cited in [X] and later correlated in the reference section by the number.
- It's missed the chemical composition, authors must provide a surface characterization, at least using a conventional technique such as XPS. It will help to the potential enduser for this material
- The paper is well discussed and planned
- Authors must apply in a surface and compare with other typical coating agents
- Try to solve these issues and will be considered for publication
Author Response
Dear Reviewer
We would like to thank the Editor and Reviewer for carefully reviewing our manuscript and giving very helpful comments for improvement.
The mnuscript have several points must be considered previous to be considered for publication in Processes.
1. The references throw the text must be formatted according the mpdi guidelines,I mean the references must be cited in [X] and later correlated in the reference section by the number.
Response: The references have been updated following the guidelines.
2. It's missed the chemical composition, authors must provide a surface characterization, at least using a conventional technique such as XPS. It will help to the potential enduser for this material
Response: We agree with the Reviewer that it would be better to further compare the surface composition of the films using XPS or other techniques. We will consider this in our future study.
3. The paper is well discussed and planned
Response: Thanks for the favorable comments
4. Authors must apply in a surface and compare with other typical coating agents
Response: We agree that it is better to include the surface composition and compare with other coating agents. We will further analyse and compare in our future study. We believe that the current data are sufficient for publication at this stage. Thank you very much for your useful suggestion.
5. Try to solve these issues and will be considered for publication
Response: We hope that we have addressed the Reviewers’comments and have significantly improved our manuscript.
Again, we would like to thank the Editor and Reviewers for valuable and constructive comments. We hope that our revision is now satisfied the Editor and Reviewers and the manuscript is suitable for publication in the Processes.
Best regards,

Round 2
Reviewer 3 Report
Well, the manuscript has been improved
The manuscript can continue the way for publication